# Buccal Administration of a Zika Virus Vaccine Utilizing 3D-Printed Oral Dissolving Films in a Mouse Model

**DOI:** 10.3390/vaccines12070720

**Published:** 2024-06-28

**Authors:** Sarthak Shah, Parth Patel, Amarae Ferguson, Priyal Bagwe, Akanksha Kale, Emmanuel Adediran, Revanth Singh, Tanisha Arte, Dedeepya Pasupuleti, Mohammad N. Uddin, Martin D’Souza

**Affiliations:** Vaccine Nanotechnology Laboratory, Center for Drug Delivery Research, College of Pharmacy, Mercer University, Atlanta, GA 30341, USA; sarthak.modi.shah@live.mercer.edu (S.S.); parth.r.patel@live.mercer.edu (P.P.); amarae.ferguson@live.mercer.edu (A.F.); priyal.bagwe@live.mercer.edu (P.B.); akanksha.madhav.kale@live.mercer.edu (A.K.); emmanuel.adediran@live.mercer.edu (E.A.); revanth.singh.sateesh@live.mercer.edu (R.S.); tanisha.manoj.arte@live.mercer.edu (T.A.); dedeepya.pasupuleti@live.mercer.edu (D.P.); uddin_mn@mercer.edu (M.N.U.)

**Keywords:** Zika, buccal, orally dissolving films (ODFs), mucoadhesive film, Guillain–Barré syndrome (GBS), nanotechnology, microparticles

## Abstract

Over the years, research regarding the Zika virus has been steadily increasing. Early immunization for ZIKV is a priority for preventing complications such as microencephaly and Guillain–Barré syndrome (GBS). Unlike traditional vaccination approaches, oral dissolving films (ODFs) or mucoadhesive film technology is an emerging, exciting concept that can be used in the field of pharmaceuticals for vaccine design and formulation development. This attractive and novel method can help patients who suffer from dysphagia as a complication of a disease or syndrome. In this study, we investigated a microparticulate Zika vaccine administered via the buccal route with the help of thin films or oral dissolving films (ODFs) with a prime dose and two booster doses two weeks apart. In vitro, the ODFs displayed excellent physiochemical properties, indicating that the films were good carriers for vaccine microparticles and biocompatible with the buccal mucosa. In vivo results revealed robust humoral (IgG, subtypes IgG1 and IgG2a) and T-cell responses (CD4+/CD8+) for ZIKV-specific immunity. Both the Zika MP vaccine and the adjuvanted Zika MP vaccine affected memory (CD45R/CD27) and intracellular cytokine (TNF-α and IL-6) expression. In this study, ZIKV vaccination via the buccal route with the aid of ODFs demonstrated great promise for the development of pain-free vaccines for infectious diseases.

## 1. Introduction

Zika virus (ZIKV), belonging to the Flaviviridae family, was first identified in a Ugandan forest in 1947, with the initial human cases documented in Tanzania and Uganda in 1952 [1,2,3,4,5]. Subsequent outbreaks occurred in Micronesia in 2007 and in New Caledonia and Polynesia from 2013 to 2014, with over 80 countries now reporting its presence. The significant outbreak in Brazil, with over 100,000 reported cases of Zika, as well as its spread in South America, drew attention from the research community and raised public awareness regarding the congenital defects that had previously been considered anecdotal during the 2013 and 2014 epidemic in the Pacific Ocean islands [1,2]. Pregnancy complications range from congenital defects like microencephaly to delayed brain development and hormonal imbalances, while in adults, ZIKV has been linked to Guillain–Barré syndrome, which affects the peripheral nervous system [2,6]. ZIKV transmission extends beyond mosquito bites to include congenital, sexual, and intranasal routes [5]. Another interesting complication is that this virus can cause several dental issues or issues related to the oral mucosa, such as blisters on the tongue and/or around the gums [7,8,9]. Recent studies highlight sexual transmission, with ZIKV found in various bodily fluids. Transmission via oronasal secretions has been demonstrated in animal models, suggesting potential human infection routes beyond traditional vectors [5].

Since mucosa-associated lymphoid tissue (MALT) is located in the oral cavity, the buccal administration route is a highly appealing method for delivering vaccines to stimulate mucosal immunity. While oral, nasal, and pulmonary vaccine delivery methods have been extensively discussed, the lesser-known sublingual and buccal routes have received relatively less attention [10]. Mucosal tissues, owing to their expansive surface area and immunological proficiency, present appealing options for vaccine administration and targeting [10]. The buccal mucosa, rich in antigen-presenting cells, offers a patient-friendly, pain-free alternative, likely increasing vaccination rates [10]. Once the ODF is placed in the buccal mucosa, APCs called Langerhans cells and M-cell-like cells can take up the Zika vaccine MPs (Figure 1). Once APCs uptake and have successfully processed the Zika MPs, the APCs can travel to the lymph nodes to initiate the interaction with T cells to mount a strong immune response. We plan to test a microparticle formulation encapsulating the whole inactivated Zika virus in polymeric MP, which will preserve the antigenic epitopes present on the virus. Lyophilized MPs have several advantages, such as increased shelf life which reduces the need for cold chain storage equipment [1,2]. Our microparticulate (MP) vaccine, encapsulating the inactivated Zika virus, was embedded in a 3D-printed orally dissolving film (ODF) for buccal delivery. Orally dissolving film (ODF) for buccal delivery provides a large surface area, approximately 50 cm^2^, for disintegration and immune response stimulation [10]. Due to the accessibility of the buccal mucosa and the presence of resident dendritic cells, the buccal mucosa is an ideal site for vaccination because it elicits mucosal and systemic immune responses [5]. By utilizing a fully automated 3D printing process for ODF fabrication, our cost-efficient method eliminates manual labor, ensuring reproducibility. This innovative approach holds promise for improving vaccine accessibility, reducing costs, and simplifying the vaccination process. To address vaccine hesitancy stemming from pain associated with intramuscular or subcutaneous administration, we propose a pain-free approach [11]. This integrated approach addresses the limitations of traditional injection methods and holds promise for developing an effective Zika vaccine. Combining polymeric particulate vaccine delivery with buccal immunization with the aid of ODFs can yield an effective Zika vaccine, as demonstrated with the measles vaccine in porcine pigs, ensuring robust humoral and cellular immune responses [11].

## 2. Materials and Methods

### 2.1. Materials

The Zika virus strain PRVABC59, with a viral titer of 1 × 10^6^ PFU/mL, was kindly provided by the Centers for Disease Control and Prevention (CDC) in Colorado [1]. Adjuvants, the FDA-approved Alhydrogel^®^ and Monophosphoryl Lipid A (MPL-A^®^), were obtained from InvivoGen in San Diego, CA, USA. The polymer poly(D,L-lactide-co-glycolide) grade 75:25 (PLGA) (Resomer^®^ RG 752 H) was acquired from the Evonik Corporation in Birmingham, AL, USA. Cryoprotectant, Trehalose dihydrate, and poly-vinyl alcohol (PVA) (with a molecular weight ranging from 30,000 to 70,000) were purchased from Sigma-Aldrich in St. Louis, MO, USA. Swiss Webster mice (4–6 weeks old, female) for in vivo studies were procured from Charles River Laboratories in Wilmington, MA, USA. Goat anti-mouse secondary IgG, IgG1, IgG2a, IgG3, IgA, and IgM, conjugated to Horseradish Peroxidase (HRP) for ELISA experiments, were purchased from Invitrogen™ in Rockford, IL, USA. For flow cytometry analysis, Allophycocyanin (APC)-labeled anti-mouse CD4 helper antibody and fluorescein isothiocyanate (FITC)-labeled anti-mouse CD8a antibody were obtained from Invitrogen™, Thermo Fisher Scientific in Waltham, MA, USA. Oral dissolving films (ODFs) were made using the INKREDIBLE plus^®^ 3D bioprinter which was acquired from CELLINK in Gothenburg, Sweden. Film-forming polymers Kollidon F90, Kollidon VA64, Kollidon K90, and Kollidon K64 were purchased from BASF in Houston, TX, USA.

### 2.2. Methods

#### 2.2.1. Formulation of the Microparticulate Vaccine

The microparticle formulation process has been previously reported by our group [1,2,12]. In brief, vaccine microparticles were prepared using the double emulsion solvent evaporation technique to encapsulate the inactivated Zika virus within a poly(D,L-lactide-co-glycolide) (PLGA) polymer matrix [1]. Initially, the inactivated Zika antigen was emulsified in a 2% polymer solution (PLGA in dichloromethane (DCM)) employing an Omni THQ probe homogenizer (Kennesaw, GA, USA). This step produced the primary emulsion. Subsequently, the primary emulsion was further emulsified with a 0.1% polyvinyl alcohol (PVA, MW 30,000–70,000, Sigma-Aldrich) solution, resulting in a second or double emulsion. To reduce the microparticle size, the final emulsion underwent six cycles of processing through a Nano DeBEE high-pressure homogenizer. Following homogenization, the emulsion was stirred for 4 h to allow DCM evaporation. After evaporation, the emulsion was subjected to ultracentrifugation to concentrate the microparticles. Upon ultracentrifugation, the concentrate was lyophilized with the addition of trehalose as a cryoprotectant, using a FreeZone Triad benchtop freeze dryer (Labconco™, Kansas City, MO, USA). Two microparticulate adjuvants, microparticles encapsulating Alhydrogel^®^ and MPL-A^®^, were formulated using a methodology akin to the one described above.

#### 2.2.2. Preparation of 3D-Printed Orally Dissolving Films (ODFs)

The formulation of ODFs involves the use of various biodegradable polymers. First, Kollidon 90F (16.24% *w*/*v*) and Kollidon VA64 (1.06% *w*/*v*) were prepared using ethanol as the solvent. Next, PEG 2000 (0.6% *w*/*v*) was prepared and slowly added to the previous mixture. This mixture was covered with foil and stirred for 4 h. The resultant mixture, along with vaccine microparticles, was loaded onto the nozzle head of the CELLINK INKREDIBLE plus^®^ 3D bioprinter (Gothenburg, Sweden). G-code was created for printing the ODFs. The ODF formulation was printed in an in-lab-prepared mold. This mold was prepared using a SYLGARD 186 silicone elastomer kit for two polymers, and the resulting mixture was placed onto the lid of a 96-well plate to make the mold. The resulting mixture, along with inactivated Zika microparticles, was loaded into a 3D printer machine set at room temperature (RT) and at 50 kPa (pressure). Once the 3D printer was finished, the 96-well plate was placed in the heating chamber for 5 min at 4 °C and then placed in the desiccator overnight.

#### 2.2.3. Characterization of Vaccine Particles: Particle Size, Zeta Potential, and Laser Particle Counter

The Zetasizer Nano ZS (Malvern Pananalytical, Westborough, MA, USA) was used to assess both the particle size and zeta potential [12]. Specifically, a 12 mm square polystyrene cuvette (DTS0012) was utilized for sizing the particles, while a folded capillary cell (DTS1070) was employed to determine the zeta potential or particle charge. The sample comprised a uniform suspension of microparticles (MPs) at a concentration of 0.05 mg/mL, and the experiment was replicated three times. Additionally, a laser particle counter (Spectrex PC-2200, Redwood City, CA, USA) was employed to enumerate the Zika microparticles present in 1 mL of phosphate-buffered saline (PBS). For this purpose, precisely 2 mg of ZIKV MPs was weighed and dispersed in 2 mL of PBS. These measurements were repeated eight times (n = 8).

#### 2.2.4. Morphological Characterization of Oral Dissolving Films (ODFs)

Instrument Phenom™ (Nanoscience instruments, Phoenix, AZ, USA) was used to obtain scanning electron microscopy (SEM) of microparticles-loaded ODFs [2,13,14]. Briefly, the Zika MP-ODF or adjuvanted Zika MP-ODF was loaded on a single stub using a double-coated carbon conductive PELCO Image Tabs™ (Ted Pella Inc., Redding, CA, USA) and analyzed.

#### 2.2.5. Fourier Transform Infrared Microscopy (FTIR)

Vaccine microparticles were analyzed using Fourier transform infrared (FTIR) spectroscopy (Shimadzu IRAffinity-1S; Tampa, FL, USA). The PLGA polymers, blanks (no vaccine MPs), ODFs, and Zika MP-loaded ODFs were measured. Briefly, 1 mg of the PLGA polymer was placed on a ZnSe crystal puck and analyzed. Next, the blank (no vaccine) and Zika MP vaccine ODFs were placed on the puck, and the spectra were measured. This experiment was repeated three times, and the spectra of each sample were determined.

#### 2.2.6. Physiochemical Assessment of the ODFs

Average film thickness: The thickness of the prepared films was measured using a Mitutoyo electronic digital caliper (Japan) [13,14]. The thickness of each film was measured at 8 different points, and the average thickness was calculated. The final average thickness was recorded in mm.

Weight variation and diameter: Each ODF was carefully placed on an electronic weighing balance to determine its weight. Six (n = 6) ODFs were weighed: blank ODFs (no vaccine MPs), Zika MP ODFs, and Zika + Adjuvants (Alhydrogel and MPL-A) MP ODFs. The average weights of the films and their standard deviations were calculated. The final average weight was recorded in mg. In addition, the diameter of each ODF was determined using a Vernier caliper by measuring through the center. The results of each ODF were recorded as the mean ± SD.

Disintegration test of the ODF formulation [13]: Each ODF formulation film strip was placed in a Petri dish (141 cm^2^). Then, 3 mL of PBS solution was added to each formulation. The petri dish was then placed on a shaker to allow the film strip to completely disintegrate. The time required for the film to completely dissolve in the solution was recorded from 3 parallel experiments.

Surface pH of the ODF formulation [13,15,16]: Each ODF formulation was placed in 2 mL of artificial saliva. The surface pH was noted by placing the pH electrode directly on top of the ODF. The pH of the artificial saliva was measured separately by placing the electrode only in artificial saliva in a small beaker. The experiment was completed on four ODFs for each formulation and the average pHs and SDs were calculated.

Stability of the ODF formulation: In our films, stability testing was performed as per International Council on Harmonization (ICH) guidelines under several conditions. Four film strips from each ODF formulation (blank ODF, Zika MP vaccine ODF, and adjuvanted Zika MP ODF) were wrapped with butter paper and then with aluminum foil to be stored in the stability chamber. These films were stored under two storage conditions: 25 °C/60% relative humidity (RH) (Zone II) and 40 °C/75% R.H. (zone IVb). Stored film strips were sampled at varying intervals (0, 1 week, 1 month, 3 months, 6 months, and 12 months) for further testing for overall appearance, physiochemical properties, and weight variation.

#### 2.2.7. Evaluation of Immunostimulatory Potential of Vaccine MPs Loaded in ODFs

The in vitro immunostimulatory potential of Zika vaccine microparticles with or without adjuvant (alum and MPL-A) microparticles was assessed by measuring nitric oxide released by DC 2.4 dendritic cells using the Griess assay. This assay measures nitric oxide, an important innate immune marker that is released after stimulation of DC cells and which can be quantified [1,2,12]. Briefly, cells were plated in a 96-well plate with a seeding density of 1 × 10^4^ cells/well. The DCs in different wells were exposed to various treatments. Treatment groups: Unstimulated cells were used as controls; lipopolysaccharide (LPS) was used as a positive control; and blank ODF (no vaccine), Zika MP ODF, and adjuvanted Zika MP ODF were used as controls. Each ODF sample was prepared by dissolving the ODF in 1 mL of EMEM. After the various treatment groups were added, the cells were placed in an incubator for 24 h. After a 24 h incubation period, the culture supernatants from each treatment group were collected and treated with the Griess reagent system, which includes 1% sulfanilamide in 5% phosphoric acid and 0.1% N-1-naphthyl ethylenediamine dihydrochloride (NED), sourced from Fischer Scientific in Hampton, NH. This process is crucial as it converts nitric oxide into nitrite, leading to the formation of a pink-colored azo compound. Subsequently, the Griess reagent system-treated samples were incubated for 3 h in a 96-well plate within an incubator. Following incubation, the plate was transferred to a BioTek^®^ Synergy H1 microplate reader from BIO-TEK Instruments in Winooski, VT, USA, where absorbance readings were taken at 540 nm. The experiment was conducted in triplicate. The concentration of nitrite was determined utilizing a standard curve of sodium nitrite, obtained from Fischer Scientific in Hampton, NH, USA, with standard concentrations ranging from 3 µM to 200 µM.

#### 2.2.8. Evaluation of the Ability of the MP Vaccine to Induce Autophagosomes

An autophagy investigation was conducted to evaluate the ability of the MP vaccine to induce autophagosomes. The experiment was conducted using dendritic cells (DC 2.4) stably transfected with the autophagy marker GFP-LC3 [12]. These DC cells were cultured and plated onto a 12-well plate for overnight incubation. After 24 h, the Dulbecco’s Modified Eagle Medium (DMEM) was removed, and the plate underwent three washes with PBS. Subsequently, the DCs were exposed to various treatments, including positive and negative controls (no exposure and rapamycin inhibitor, respectively), Zika solution (50 µg/well), Zika MP ODF (50 µg/well), and adjuvanted Zika MP ODF (50 µg/well Zika MP + 25 µg/well adjuvant), for 24 h. Each sample of oral dissolving film (ODF) was dissolved in 1 mL of DMEM, vortexed, and added to each well. The plate was then incubated overnight at 37 °C with 5% CO_2_. After this incubation period, the supernatant was removed, and the cells were washed carefully with PBS six times. Following the washes, the cells were fixed using a 4% paraformaldehyde solution. Subsequently, the fixed cells were stained with DAPI (1 mg/mL), a nuclear stain from Thermo Fisher Scientific in Rockford, IL, USA, for 10 min. Autophagy was observed using the live cell imager Biotek (Lionheart/FX, Winooski, VT, USA). Additionally, the BD Accuri C6 Plus flow cytometer from BD Bioscience in San Jose, CA, USA, was utilized for the quantitative assessment of autophagosomes [2].

#### 2.2.9. Evaluation of the Cytotoxicity of Vaccine MPs Loaded in ODFs

MTT assay was performed to assess the cytotoxicity of the vaccine MPs in vitro [2,12]. The cytotoxicity of blank MP ODF (no vaccine), Zika MP ODF, and adjuvanted Zika MP ODF was assessed using the 3-(4,5-dimethylthiazol-2-yl)-2,5-diphenyl tetrazolium bromide (MTT) cell viability assay. In brief, dendritic 2.4 cells (DC) were seeded in a 96-well plate at a density of 1 × 10^4^ cells/well. Negative and positive controls consisted of cells only and dimethyl sulfoxide (DMSO) groups, respectively. The DC cells were treated with blank MP ODF (no vaccine) (50 μg/well), Zika MP ODF (50 μg/well), and adjuvanted Zika MP ODF (50 μg/well + 25 μg/well), for 24 h. Following the 24 h period, the cells were washed three times to eliminate extracellular particles. Subsequently, the 96-well plate was incubated with MTT reagent (5 mg/mL in PBS) for 2 h, followed by the addition of 50 µL of DMSO to dissolve the formazan precipitate. The plate was covered with foil and then shaken at room temperature for 25 min. Absorbance was measured at 570 nm using a microplate reader (BioTek^®^ Synergy H1, BIO-TEK Instruments, Winooski, VT, USA). Cell viability was calculated as a percentage relative to cells treated with growth media only.

#### 2.2.10. In Vivo Study Design for Buccal Immunization with Oral Dissolving Film (ODF)

Approval for all animal experiments was obtained from the Mercer University Institutional Animal Care and Use Committee (IACUC) protocol (#A2303001). Swiss Webster (SW) mice, 4–6 weeks old, were obtained from Charles River Laboratories (Wilmington, MA, USA). Mice were randomly distributed and received a prime dose along with two booster doses, as shown in Figure 2, via buccal administration using oral dissolving films (ODFs). Serum was collected biweekly to determine antibody response for total IgG, IgA, and IgG subtypes (IgG2a, IgG1, and IgG3) via enzyme-linked immunosorbent assay (ELISA). Post-challenge, immune organs (spleen, inguinal, and axillary lymph nodes) were harvested and analyzed later for flow cytometry (FACS).

#### 2.2.11. Measurement of Zika-Specific Antibody Titers Using ELISA

Enzyme-linked immunosorbent assay (ELISA) was conducted to detect several antibodies: total IgG, IgA, and IgG subtypes (IgG2a, IgG1, and IgG3) [1,2]. Initially, ELISA plates (Microlon^®^, Greiner Bio-One, North Carolina USA) were coated with inactivated Zika strain PRVABC59 (50 ng/well) followed by blocking with 3% BSA (37 °C, 2 h) the next day. Diluted serum samples were added, and the plate was incubated for 1 h. Next, HRP-conjugated goat anti-mouse secondary antibody was added and incubated for 2 h. 3,3′,5,5′-tetramethylbenzidine (TMB) (BioLegend^®^, San Diego, CA, USA) was used as a substrate and 0.3 M sulfuric acid was used to halt the reaction. Plates were washed three times (0.1% Tween 20 in PBS) at every step. The plate was measured using a BioTek^®^ Synergy H1 microplate reader (BIO-TEK Instruments, Winooski, VT, USA) at 450 nm.

#### 2.2.12. Measurement of Cellular, Memory, and Cytokine Responses

Immune organs such as the spleen, inguinal, and axillary lymph nodes were cryopreserved at −80 °C with 5% *v*/*v* dimethyl sulfoxide (DMSO) as a cryoprotectant after sacrifice to determine CD4, CD8, memory markers, and intracellular cytokines. All samples were analyzed using a BD Accuri C6 Plus flow cytometer (BD Bioscience, San Jose, CA, USA). Following the thawing of cell suspension, cells were stimulated with 5 µg/mL IL-2 and incubated overnight. The next day, 100 µL of anti-mouse APC-labeled CD4 and FITC-labeled CD8a markers in PBS was added to the cell suspension followed by incubation for 1 h. Cells were washed after the incubation period three times. Cytokine markers (TNF-α and IL-6) and memory response markers (CD27 and CD45R) were also analyzed. The unstained live cell population was gated during flow cytometry and a total of 5000 events were counted for each sample.

#### 2.2.13. Statistical Analysis

Statistical analyses were performed using GraphPad Prism, version 10.2.3 (GraphPad Software, San Diego, CA, USA, https://www.GraphPad.com (accessed on 10 May 2024)). Unless stated otherwise, all experiments were conducted in triplicate. Initially, normality was assessed using the Shapiro–Wilk test, while variance of the data was evaluated using the Brown–Forsythe test. For normally distributed data, a one-way analysis of variance (ANOVA) with Tukey’s post hoc test was applied. For data not conforming to a normal distribution, a nonparametric Kruskal–Wallis test followed by a post hoc analytical test was employed. Statistical significance was considered at *p* values < 0.05 in all cases. Unless specified otherwise, data are presented as the mean ± standard error of the mean (SEM).

## 3. Results

### 3.1. Vaccine Microparticle Characterization: Particle Size, Zeta Potential, and Laser Particle Counter

The formulation of Zika vaccine microparticles utilized a double emulsion technique, following methodologies previously outlined by our laboratory [2,12]. We assessed each MP formulation with and without an ODF to determine whether the ODF caused any changes in the physiochemical profile of the microparticles in terms of the average particle size (nm), polydispersity index, and zeta potential. Figure 3A shows blank MP (no vaccine), Zika MP vaccine, Alhydrogel (alum) MP, and MPL-A MP samples. In Figure 3B, each ODF formulation was dissolved in 1 mL of PBS, and the following OFs were measured: blank MP (no vaccine), ODF, Zika MP vaccine, ODF, Alhydrogel (alum) MP vaccine, ODF, and MPL-A MP vaccine. The particle size decreased within the range of 400–1000 nm with and without ODFs. The observed low polydispersity index indicated a uniform size distribution of the particles with and without the ODF formulation. Furthermore, the negative surface charge of blank MP (no vaccine), the Zika MP vaccine, the Alhydrogel (alum) MP, and the MPL-A MP suggested the absence of particle aggregation with and without ODFs.

### 3.2. Morphology of ODFs by Scanning Electron Microscopy (SEM)

Scanning electron microscopy (SEM) was used to image ODF for adjuvanted Zika MP ODF and Zika MP ODF (Figure 4A). Adjuvanted Zika MP ODF and Zika MP ODF SEM images show the successful incorporation of vaccine MPs into the ODF. Quantitative autophagy results are shown in Figure 4B. There was a significant percentage of expression of autophagy or autophagosomes for both adjuvanted and unadjuvanted Zika MP ODFs compared to the no-treatment group.

### 3.3. Fourier Transform Infrared Microscopy (FTIR)

Fourier transform infrared (FTIR) spectroscopy was utilized to investigate the incorporation of the inactivated Zika antigen in the polymer poly(D,L-lactide-co-glycolide) grade 75:25 (PLGA). Second, FTIR was utilized to assess the incorporation of ZIKV MPs in ODFs. Figure 5 shows the FTIR spectra of the Zika MPs, blank ODFs (no vaccine), and Zika MP-ODFs.

### 3.4. Physiochemical Assessment of the ODFs

The formulated ODFs were tested with and without vaccine microparticles. The ODFs were assessed for average weight (mg), thickness (µm), diameter (mm), disintegration time (min), and pH. A summary of the results is shown in Table 1. There was no significant weight variation among the three groups. The thickness of the ODFs ranged from 0.22 to 0.27 µm. As expected, the thickness of the ZIKV MP combined with adjuvant ODFs was greater than that of any other group. In the Zika MP ODF group, the disintegration time was 3.01 ± 1.18 min, indicating that the Zika MPs can be released from the multipolymer matrix within 3–4 min. Similarly, the Zika MP combined with two adjuvants (Alum and MPLA MPs) ODF group showed a rapid disintegration time of 3.39 ± 1.26 min. The pH of the buccal mucosa was approximately 6.5–7.2, which is suitable for the buccal mucosa. Physiochemical assessment of blank ODF (without vaccine), Zika MP vaccine ODF, and adjuvated Zika MP vaccine ODF was performed after 1 week, 3 months, 6 months, and 1 year. The avg. weight, thickness, disintegration time, and pH did not change significantly from 1 week to 1 year for all the ODF formulations.

**Table 1 vaccines-12-00720-t001:** Physiochemical characterization of ODFs: average weight, thickness, diameter, disintegration time, and pH with and without in ODF formulation.

	Without Vaccine	With Zika Vaccine	
	Blank ODF	Zika MP ODF	Adjuvanted Zika MP ODF
Avg. Weight (mg)	8.83 ± 0.89	8.98 ± 0.87	10.0 ± 1.37
Thickness (µm)	0.22 ± 0.11	0.25 ± 0.09	0.27 ± 0.07
Diameter (mm)	0.40 ± 0.01	0.40 ± 0.01	0.40 ± 0.01
Disintegration time (min)	4.00 ± 0.52	3.01 ± 1.18	3.39 ±1.26
pH	7.25 ± 0.07	7.05 ± 0.20	7.19 ± 0.12

**Table 2 vaccines-12-00720-t002:** Physiochemical characterization of blank ODFs after 1 week, 3 months, 6 months, and 1 year for average weight, thickness, disintegration time, and pH.

		Blank ODF		
	1 Week	3 Months	6 Months	1 Year
Avg. Weight (mg)	8.03 ± 01.09	8.08 ± 0.87	8.25 ± 0.45	8.32 ± 02.10
Thickness (µm)	0.23 ± 0.10	0.25 ± 0.09	0.28 ± 0.11	0.27 ± 0.15
Disintegration time (min)	3.95 ± 0.43	4.01 ± 0.43	3.86 ± 0.32	3.23 ± 0.17
pH	7.15 ± 0.03	7.18 ± 0.10	7.20 ± 0.01	7.23 ± 0.16

**Table 3 vaccines-12-00720-t003:** Physiochemical characterization of Zika MP vaccine ODFs after 1 week, 3 months, 6 months, and 1 year for average weight, thickness, disintegration time, and pH.

		Zika MP Vaccine ODF		
	1 Week	3 Months	6 Months	1 Year
Avg. Weight (mg)	8.76 ± 0.47	8.56 ± 0.23	8.66 ± 0.43	8.23 ± 0.33
Thickness (µm)	0.26 ± 0.10	0.27 ± 0.04	0.27 ± 0.12	0.26 ± 0.02
Disintegration time (min)	3.05 ± 0.06	3.12 ± 0.23	3.23 ±0.11	3.15 ±0.15
pH	7.08 ± 0.10	7.00 ± 0.11	7.04 ± 0.10	7.03 ± 0.05

**Table 4 vaccines-12-00720-t004:** Physiochemical characterization of adjuvanted Zika MP vaccine ODFs after 1 week, 3 months, 6 months, and 1 year for average weight, thickness, disintegration time, and pH.

	Adjuvanted Zika MP Vaccine ODF			
	1 Week	3 Months	6 Months	1 Year
Avg. Weight (mg)	9.8 ± 0.32	9.98 ± 0.82	10.1 ± 0.23	9.8 ± 0.32
Thickness (µm)	0.26 ± 0.09	0.25 ± 0.11	0.28 ± 0.12	0.29 ± 0.15
Disintegration time (min)	3.42 ±0.45	3.54 ± 0.32	3.16 ±1.16	3.40 ± 0.21
pH	7.23 ± 0.10	7.15 ± 0.13	7.12 ± 0.09	7.22 ± 0.12

### 3.5. Immunostimulatory Potential and Cytotoxicity Profile of ODFs

The Griess assay measured ZIKV vaccine microparticle immunostimulatory potential in vitro [2,12]. Nitric oxide released by dendritic cells 2.4 (DC) was quantified for blank ODFs, Zika MP vaccine ODF, and adjuvanted Zika MP vaccine ODF, with cells only and Lipopolysaccharide (LPS) as controls (Figure 6A). Both vaccine groups showed significantly higher NO release than untreated cells. Vaccine microparticles’ cytotoxicity was assessed using the MTT assay (Figure 6B) [1]. DCs treated with different treatment groups for 24 h showed no cytotoxic effects. Negative and positive controls were cells only and DMSO. The formulated vaccine microparticles in the ODFs were non-cytotoxic.

### 3.6. The Ability of Zika Vaccine MP-ODFs to Induce Autophagosomes

Autophagy is a sophisticated process for the formation of double-membrane vesicles called autophagosomes. Autophagy is essential for antigen cell presentation via major histocompatibility complex I or II. Once the antigen is processed and presented on MHC complexes (MHCs) I and II, this can alert CD4+ and CD8+ T cells in response to an infection [12]. In this experiment, there were five groups: the no-treatment group (dendritic cells only), the inhibitor group (rapamycin), the Zika solution group, the Zika MP ODF group, and the adjuvanted Zika MP ODF group (Figure 7). Green dye is specific for autophagosomes. This green dye is a fluorescent protein-conjugated light chain 3 (GFP-LC3 puncta) that represents active autophagy. The adjuvant Zika MP ODFs showed significantly greater numbers of autophagosomes than the Zika solution (****, *p* < 0.0001). The results of the autophagy analysis with a live cell imager (BioTek) are shown in Figure 7.

### 3.7. Zika-Specific Humoral Antibodies

The humoral response was assessed in Swiss Webster mice following buccal administration using ODFs. Mice received a prime dose followed by two booster doses administered two weeks apart (at week 2 and week 4). Enzyme-linked immunosorbent assay (ELISA) was used to analyze the serum samples that were collected biweekly [1,2]. IgG antibodies, being the most abundant in serum and crucial for long-term immunity against infections, were of particular interest. Figure 8 depicts the Zika-specific IgG titers observed in Swiss Webster mice. The ZIKV vaccine and the unadjuvanted ZIKV vaccine groups exhibited robust total IgG antibody responses. IgG titers were significantly higher from weeks 2 to 11 compared to the untreated group. Notably, IgG titers remained significantly elevated in both adjuvanted and unadjuvanted vaccinated groups even after the challenge at week 10. The adjuvanted Zika MP vaccine induced a robust IgG antibody response at weeks 2, 4, 6, 8, and 11 compared to mice receiving no vaccine. In addition, other antibodies were measured, including IgA and IgG subtypes (IgG1, IgG2a, and IgG3).

One of the objectives in formulating a buccal Zika MP vaccine was to ascertain the presence of IgA antibodies. The presence of mucosal IgA antibodies elicited through the buccal route can play a critical role in neutralizing the virus at a potential site of entry. Figure 9 shows the robust IgA titer detected postvaccination. Our findings demonstrate significant and sustained production of IgA antibodies over the long term. Evaluation of IgA antibodies revealed that the adjuvanted Zika MP vaccine ODF induced robust IgA antibody levels at weeks 2, 4, and 6, which decreased at week 8 but increased post-challenge at week 11. Both adjuvanted and unadjuvanted Zika MP vaccine ODF formulations elicited significantly greater IgA titers than the no-treatment group.

To elucidate if the Zika MP vaccine elicited a Th1- or Th2-mediated immune response, ELISA was performed to measure IgG2a and IgG1 antibodies. IgG subtypes IgG1 (Figure 10), IgG2a (Figure 11), and IgG3 (Figure 12) were evaluated [17,18,19,20]. IgG1 levels peaked at week 11, indicating a robust Th2 response, notably higher with adjuvanted Zika MP vaccine ODF compared to no treatment across all weeks. Conversely, IgG2a levels, indicative of a Th1 response (Figure 11), remained significantly elevated from week 2 to week 11 in both vaccinated groups compared to controls. Post-challenge at week 11, IgG2a levels remained significantly higher, demonstrating the ability of our ODF-formulated vaccine MPs to induce a Th1 response. Regarding IgG3 antibodies, pivotal in infectious diseases for bolstering viral control, activating complement, and enhancing antibody-dependent cellular cytotoxicity (ADCC) responses [21], we noted a peak at week 4, succeeded by a decline from weeks 6 to 11 (Figure 12). The vaccine demonstrated notably higher IgG3 subtype titers compared to the no-treatment group.

### 3.8. Zika-Specific Cellular T-Cell Responses Following Buccal Immunization with ODFs

In lymph nodes, both Zika MP vaccine ODF and adjuvanted Zika MP vaccine ODF increased CD4+ helper T cells and CD8+ cytotoxic T cells. Figure 13 shows CD4+ (A) and CD8+ (B) T-cell markers in lymph nodes, with significantly higher CD4+ cells in the adjuvanted group (A) than the non-adjuvanted (B). Both vaccine groups had a higher percentage of cell counts than the untreated group. In the spleen, the vaccine induced fewer CD4+ (C) than CD8+ (D) T cells, with the adjuvanted vaccine significantly increasing both compared to untreated cells.

### 3.9. Memory and Intracellular Cytokine Response after Buccal Vaccination with ODFs

To evaluate the memory response triggered by our Zika MP vaccine, we examined CD45R and CD27 markers in the spleen and lymph nodes (Figure 14). Robust expression of memory B cell markers was observed in both immune organs. In lymph nodes, more significant levels of CD45R and CD27 markers were induced (Figure 14A,B) by the vaccine, with and without adjuvants, compared to the no-treatment group. The adjuvanted vaccine displayed a higher CD27 count percentage compared to the unadjuvanted vaccine, although CD45R expression did not significantly differ between them (Figure 14B). In spleen tissue, both CD45R and CD27 markers showed a significant memory response (Figure 14C,D), with adjuvanted and unadjuvanted vaccines inducing higher responses than the no-treatment group. Notably, CD27 expression was prominently high in both spleen and lymph nodes (Figure 14B,D), while CD45R expression was significantly elevated in both tissues (Figure 14A,C). However, the expression of CD45R in both the adjuvanted and unadjuvanted Zika MP vaccine groups was significantly higher than in the no-treatment group (Figure 14D). Figure 15 shows the Zika-specific expression of intracellular markers in the spleen. In the spleen, the expression of intracellular cytokines IL-6 (A) and TNF-α (B) was notably elevated compared to the group receiving no treatment. Specifically, the expression of IL-6 was significantly higher, with levels approximately fourfold greater than those of TNF-α. Furthermore, the adjuvanted Zika MP vaccine ODF elicited significantly higher levels of both IL-6 and TNF-α cytokines compared to both the untreated and naïve groups.

## 4. Discussion

Although several clinical trials evaluating safety and tolerability have been completed, there is still no vaccine or treatment available for the Zika virus [22,23]. Most of the research in the development of Zika continues to use the traditional intramuscular route for vaccination. A novel alternative solution is the development of oral dissolving films (ODFs). Although there are several methods of formulating and making ODFs, they need a trained person and use an older technique called the solvent casting method. Deviating from the traditional method of vaccination encourages more patient compliance for vaccination and decreases the risk of contracting Zika [24]. In our ODF formulation, the mixture was suitable for making ODFs, as made evident by physiochemical assessments. The average diameter of the ODFs was 0.40 mm, which is a small size for inserting into a mouse in vivo model. Once the film thin is placed in the buccal cavity, an optimal ODF should allow the film-forming polymers to disintegrate quickly, releasing the Zika microparticles encapsulated in the PLGA matrix. All three groups in Table 2, Table 3 and Table 4 show a quick disintegration time. This quick disintegration time indicates that the polymers chosen to form the film matrix are suitable for a quick delivery of the Zika MP vaccine. The pH of the three groups was in the neutral range, indicating that the ODF formulation is biocompatible with the buccal mucosa.

Once the mixture or base of the ODFs was formulated, the vaccine microparticles were loaded into the thin films. One vital question was whether the ODF formulation causes any changes in the physiochemical profile of microparticles in terms of the average particle size (nm), polydispersity index, and zeta potential. The size of the vaccine particles was less than suitable for effective uptake by antigen-presenting cells (APCs). FTIR showed that Zika MPs were successfully loaded onto the ODFs, as indicated by the disappearance of 2852 cm^−1^ and 1082 cm^−1^ in the Zika MPs and the appearance of a peak at 2985 cm^−1^ (for the Zika MP-loaded ODFs), suggesting successful incorporation of Zika MPs in the oral dissolving film (ODF). No major variations in the particle size, polydispersity index, or particle charge were detected before or after the addition of MPs to the ODFs. The zeta potential or particle charge may change as the pH changes; however, the pH results showed that the formulated ODFs were in the neutral range, indicating that once placed into the buccal mucosa, the ODF was biocompatible with the pH of the buccal mucosa layer.

In our research, we developed and assessed a microparticulate Zika vaccine. Particulate vaccine formulations offer several advantages [12,25,26]. Typically, they are more immunogenic than soluble antigens due to more efficient cross-presentation [25,26]. Most soluble antigens are poorly recognized, leading to limited endocytosis by antigen-presenting cells (APCs) and reduced protective immunity against pathogens. However, encapsulating or conjugating soluble antigens with biodegradable carriers can enhance recognition and uptake by APCs [26]. Particulate carriers like PLGA effectively deliver antigens, enhancing uptake by various APCs such as dendritic cells (DCs) or macrophages. Encapsulation protects antigens or inactivated Zika virus from proteolytic degradation and aids delivery to APCs [25,26]. Our prior research demonstrated that encapsulation did not alter the Zika structure [2]. Microparticles provided sustained antigen release, resulting in a robust innate and adaptive immune response [1,2]. Stability is crucial in vaccine formulation, but microparticulate formulations overcome this by avoiding the need for cold-chain storage. Marketed intramuscular vaccines are usually in liquid form, requiring equipment to maintain low temperatures to preserve shelf life [25,26]. In our stability testing, we found that even at 12 months, films retained their physiochemical properties, and films’ morphology did not change significantly. However, additional detailed stability studies of heat capacities such as DSC can be performed since this can provide more information on the stability of the ODF. In our approach, the inactivated Zika virus served as the antigen. FDA-approved adjuvants were employed to boost immunogenicity. Alhydrogel^®^ induces a Th2-mediated response, while MPL-A^®^ triggers a Th1-mediated response. We investigated a buccal vaccine for Zika using ODFs, which may help to deliver the vaccine pain-free and elicit a balanced Th1/Th2 immune response.

The Griess assay is essential for assessing the innate immune response initiation [1,2,12]. Microparticles (MPs) are taken up by antigen-presenting cells (APCs), triggering nitric oxide (NOˑ) release, quantified by its oxidation product, nitrite. We measured NOˑ release from DC 2.4 cells to evaluate Zika MP immunostimulatory potential. Upon antigen recognition, APCs like dendritic cells release NOˑ and cytokines (IL-12, TNF, IL-6, IFN-γ), enhancing the immune response. Quantifying NOˑ clarifies Zika MPs’ role in innate immunity initiation. Blank ODFs (no vaccine) showed no immunostimulatory potential, as expected, while vaccine MPs from ODFs stimulated DCs, which was evident from the NOˑ release versus the controls. After APC uptake, MPs aid in antigen processing, as seen in autophagy assays [1,12,27,28]. DCs exposed to Zika MP or adjuvanted Zika MP ODFs had significantly more autophagosomes than Zika solution or cells only, indicating that (1) vaccine MPs were successfully taken up and processed, and (2) ODF formulation incorporating the vaccine will not inhibit the autophagy process or harm the DCs, as further supported by the results shown in the cytotoxicity study.

We found excellent humoral and cellular responses after buccal immunization with the aid of orally dissolving films (ODFs). The long-term antibody IgG was significant even after 6 weeks. A key component of our vaccine strategy was to generate IgA antibodies, which are crucial for mucosal surface protection and limiting infection spread [4,29,30]. Nasal-associated lymphoid tissue (NALT) plays a pivotal role in mucosal immunity, particularly in IgA production within the nasal cavity [5]. Studies on nasal vaccines in humans have shown significant induction of IgA and IgG titers in mucosal sites like the vagina, comparable to direct intravaginal administration [5,19,20]. Our vaccine sustained robust IgA production long-term. Additionally, we examined IgG subtypes IgG1 and IgG2a [19,20]. IgG1 initiates antibody responses against viral pathogens, binds strongly to FcγR, and participates in ADCC [18,30]. IgG2a responds to polysaccharides, DNA, or RNA viruses. Zika MP ODF and adjuvanted Zika MP ODF vaccines induced Zika-specific IgG1 and IgG2a, indicating a balanced Th2 and Th1 immune response crucial for immune health. Zika vaccine, with or without adjuvants, also showed significant IgG3 titers, critical for effector functions, complement activation, neutralization, and ADCC [18,21,30].

The Zika vaccine, with or without adjuvants, induced Zika-specific cellular T-cell responses following buccal immunization using ODFs. The adjuvanted Zika MP ODF vaccine and Zika MP ODF vaccine significantly increased CD4+ helper and CD8+ cytotoxic T-cell counts in the spleen and lymph nodes. CD4+ helper T cells can differentiate into Th1 effectors, stimulate CD8+ T cells, and transform into T follicular helper (Tfh) cells, aiding antibody production against the Zika virus [1,31]. They also enhance communication among innate cells like macrophages and natural killer (NK) cells, crucial for Zika-infected cell elimination [32,33]. Memory markers CD45R and CD27 were prominently induced in the spleen and lymph nodes by both vaccine groups, suggesting potential protection against future Zika infections.

In our forthcoming research, we aim to assess neutralization titers to gauge the efficacy of the Zika vaccine in neutralizing the virus. Due to the administration of a sub-lethal dose of the live virus to the mice in this study, we did not find any significant changes in weight. In our next study, we will challenge mice with a lethal dose of live Zika virus to assess our vaccine’s neutralization ability. In our current pre-clinical study, the spleen showed detectable intracellular cytokine expression, contrasting with minimal expression in lymph nodes. Exploring other organs, including the brain, will reveal Zika’s impact on diverse cell types. We aim to analyze Zika-specific cytokines crucial for immune cells like NK cells, macrophages, and various T cells to decipher how our microparticulate vaccine functions during infection. This comprehensive approach will clarify the vaccine’s mechanism and potential efficacy in combating the Zika virus.

## 5. Conclusions

Since the global outbreak of Zika, extensive research efforts have been dedicated to combating this virus. Presently, no vaccine or treatment options are available. In our proof-of-concept investigation, we delved into a microparticulate Zika vaccine designed to safeguard the antigen while providing sustained release to bolster immune responses. Unlike conventional needle-based methods, we explored a pain-free Zika vaccine administered through buccal immunization with the help of orally dissolving films (ODFs). The exploration of pain-free alternatives may contribute to enhanced compliance and vaccination rates worldwide. Our findings suggest that the Zika MP vaccine elicited a sustained antibody response, with the cellular and memory response being augmented with the assistance of adjuvants. Moreover, consistent with findings in the existing literature, multiple vaccine doses may be necessary to establish long-term immunity and cross-protection. Through this proof-of-concept study, we demonstrated the feasibility of formulating an intranasal Zika microparticulate vaccine capable of enhancing both humoral and cellular responses.

## Figures and Tables

**Figure 1 vaccines-12-00720-f001:**
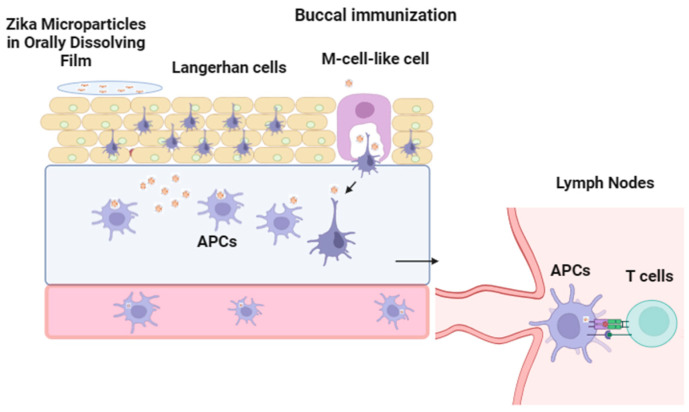
Zika MPs loaded in ODFs in the buccal mucosa. Once the Zika MPs are released from the thin film, the MPs are taken up by APCs and M-cell-like cells to process the Zika MPs. After processing, the APCs enter the blood and travel to nearby lymph nodes to initiate contact with T cells to mount a strong immune response to the Zika virus.

**Figure 2 vaccines-12-00720-f002:**
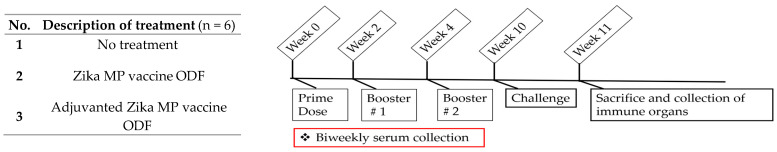
Groups of mice (n = 6) in the in vivo study (**left**). Timeline of the in vivo study (**right**) assessing the dosing status of the prime dose followed by two booster doses of the Zika MP vaccine ODF and the adjuvanted Zika MP vaccine ODF when administered via the buccal route. Challenge on Swiss Webster mice was conducted on week 10. After one week, at week 11, mice were sacrificed, and immune organs were collected.

**Figure 3 vaccines-12-00720-f003:**
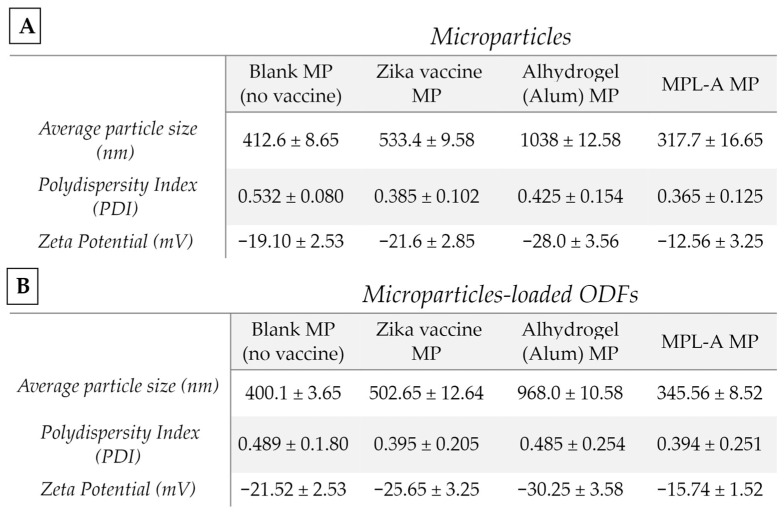
Microparticles (MPs) were formulated using a double emulsion method. MPs were characterized for percent recovery yield, particle size (nm), zeta potential (mV), polydispersity index (PDI), and number of particles/m. There was no significant difference between physicochemical characteristics of MPs versus MPs embedded in ODFs. In (**A**), Zika vaccine microparticles, Alhydrogel^®^ (Alum) MP, and MPL-A^®^ MPs were characterized. In (**B**), microparticles loaded into ODFs were characterized. The data are presented as the means ± SDs.

**Figure 4 vaccines-12-00720-f004:**
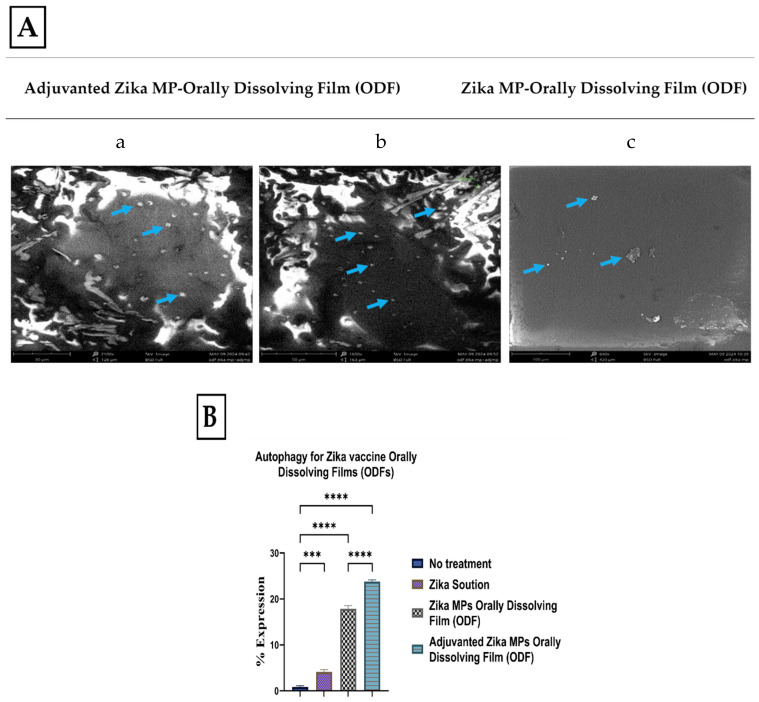
Scanning electron microscopy (SEM) images of adjuvanted Zika MP-loaded ODFs and Zika MP-loaded ODFs (**A**). Blue arrows in image (**a**–**c**) indicate presence of microparticles in ODFs. (**a**): 2100×, 5 kV, BSD full; (**b**): 1650×, 5 kV, BSD full; (**c**): 640×, 5 kV, BSD full. In (**B**), autophagy results for percentage expression for autophagosomes. Zika MP ODF and Adjuvanted ODF had a higher expression of autophagosomes than the no treatment and Zika solution. ***, *p* < 0.001; ****, *p* < 0.0001.

**Figure 5 vaccines-12-00720-f005:**
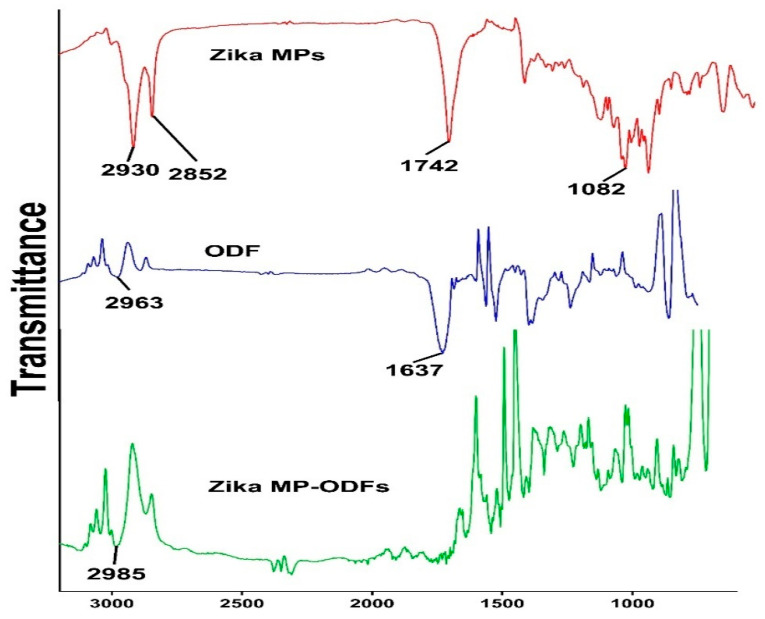
FTIR spectra of Zika MPs (red), blank ODFs (no vaccine microparticles) (blue), and Zika microparticles loaded in ODFs (green). In the Zika MP (red) FTIR spectra, several main peaks are visible at 2930 cm^−1^, 2852 cm^−1^, 1742 cm^−1^, and 1082 cm^−1^. The spectrum of the blank ODF showed two main peaks at 2963 cm^−1^ and 1637 cm^−1^. Zika MPs loaded in ODFs showed a peak at 2985 cm^−1^, but the disappearance of 2852 and 1082 was observed for Zika MPs, suggesting successful incorporation of Zika MPs in the oral dissolving films (ODFs).

**Figure 6 vaccines-12-00720-f006:**
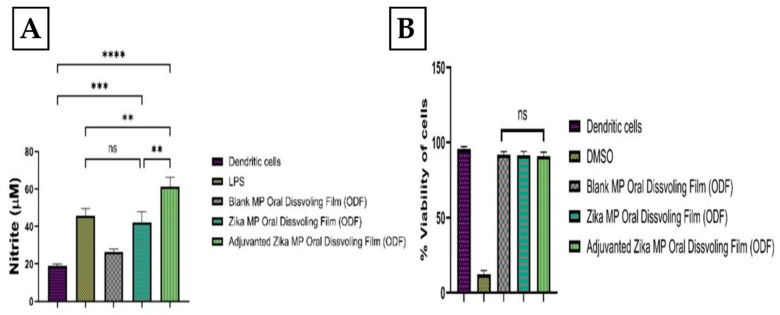
Nitric oxide production by murine dendritic cells (DC 2.4) was assessed using the Griess assay (**A**). Cells treated with Zika MP vaccine, with or without adjuvants, showed significantly increased nitric oxide compared to untreated cells or those treated with blank MP ODFs. Treatment groups included blank ODFs (50 µg/well), Zika MP vaccine ODF (50 µg/well), and adjuvanted Zika MP vaccine ODF (50 µg/well Zika + 25 µg/well adjuvant). Data are presented as means ± SEMs, analyzed using one-way ANOVA with Dunnett’s multiple comparison test (**, *p* < 0.005; ***, *p* < 0.001; ****, *p* < 0.0001). (**B**) evaluated the cytotoxicity of ODF microparticles in DC 2.4 cells using the MTT assay. Cells treated with blank ODFs, Zika MP vaccine ODF, or adjuvanted Zika MP vaccine ODF remained viable after 24 h. Data are presented as means ± SEMs, analyzed using one-way ANOVA with Dunnett’s multiple comparison test; ns, nonsignificant.

**Figure 7 vaccines-12-00720-f007:**
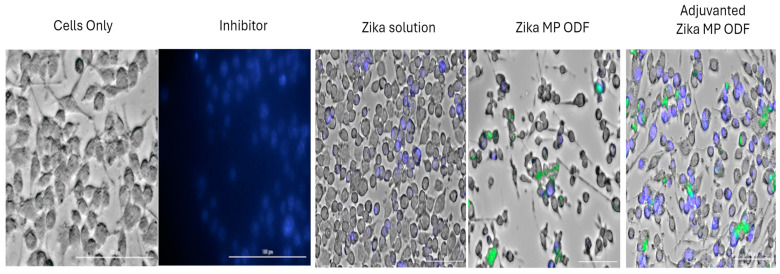
Autophagosome induction was observed using an autophagy assay. Green fluorescence protein-conjugated light chain 3 (GFP-LC3 puncta) is representative of active autophagy, while blue (DAPI, 1 mg/mL) represents nuclear staining. Cells only or Dendritic cells, rapamycin inhibitor, Zika solution (60 μg/mL), Zika MP ODF (60 μg/mL), and Zika MP ODF (30 μg/mL) + adjuvant MPs (30 μg/mL) , (images are in 100 µm). Zika MP ODF (*p* < 0.0001) and the adjuvanted Zika MP ODF had greater expression of autophagosomes than the cells treated with only the negative control or the Zika solution.

**Figure 8 vaccines-12-00720-f008:**
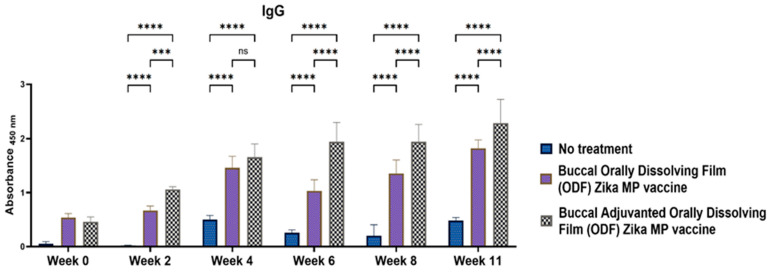
Measurement of the serum ZIKV-specific IgG titer via ELISA [1]. Animals received one prime dose at week 0 and two booster doses at weeks 2 and 4 via buccal administration. Compared with those in the untreated control group (weeks 2, 4, 6, 8, and 11), mice that received the Zika MP vaccine and the MP adjuvant via ODFs exhibited significantly greater antibody titers. Adjuvanted ZIKV vaccine MP induced significantly greater antibody titers than did the Zika MP vaccine (weeks 2, 6, 8, and 11). The data are presented as the means ± SEMs; Brown–Forsythe ANOVA test followed by Tukey’s multiple comparison test; ns, nonsignificant; ***, *p* < 0.001; ****, *p* < 0.0001.

**Figure 9 vaccines-12-00720-f009:**
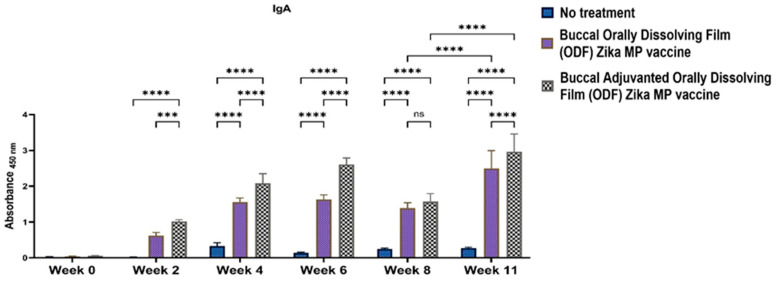
ELISA was used to analyze IgA antibodies elicited by the Zika MP vaccine ODF, with and without adjuvant [1,17]. Buccal administration of vaccine ODF, with and without adjuvants, significantly increased titers compared to untreated mice (weeks 2–6 and 11). Additionally, adjuvanted vaccine induced higher titers than non-adjuvanted vaccine (weeks 2, 4, and 11). Data represented as Mean ± SEM, Brown–Forsythe ANOVA test followed by Tukey’s multiple comparison test; ns, nonsignificant, ***, *p* < 0.001, ****, *p* < 0.0001.

**Figure 10 vaccines-12-00720-f010:**
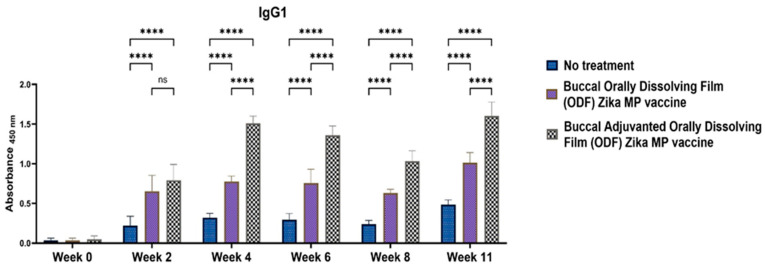
Measurement of the serum Zika-specific IgG1 titer via ELISA [1]. IgG1 antibodies are indicative of a Th2 response. Zika, adjuvanted and unadjuvanted, elicited significantly greater antibody titers than did the untreated group (weeks 4–11). Adjuvanted Zika MP vaccine ODF produced higher IgG1 antibodies than unadjuvanted vaccine (weeks 2, 4, and 11). The data are presented as the means ± SEMs; Brown–Forsythe ANOVA test followed by Tukey’s multiple comparison test; ns, nonsignificant, ****, *p* < 0.0001.

**Figure 11 vaccines-12-00720-f011:**
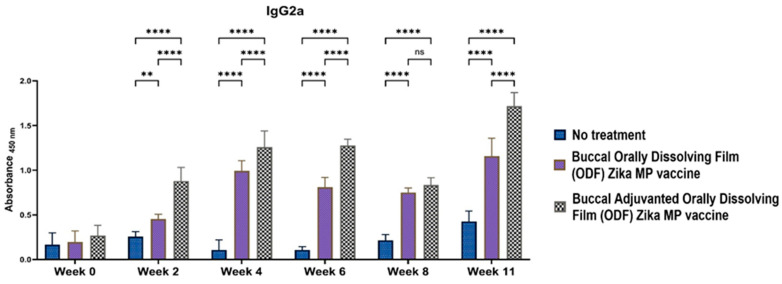
Measurement of the serum Zika-specific IgG2a titer via ELISA [1]. Mice that received the vaccine ODF, with or without adjuvants, produced significantly greater IgG2a antibody titers than did the untreated group (weeks 2–11). The adjuvanted vaccine ODF elicited higher IgG2a antibodies than the no-treatment group (weeks 2, 4, and 6 and post-challenge week 11). The data are presented as the means ± SEMs; Brown–Forsythe ANOVA test followed by Tukey’s multiple comparison test; ns, nonsignificant, **, *p* < 0.01, ****, *p* < 0.0001.

**Figure 12 vaccines-12-00720-f012:**
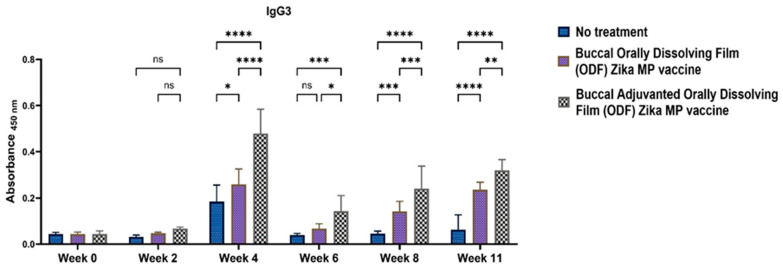
Measurement of the serum Zika-specific IgG3 titer via ELISA [1]. Mice received one prime dose at week 0 and two booster doses at weeks 2 and 4 via buccal administration. Zika MP vaccine ODF generated strong IgG3 antibodies at week 4, which then gradually decreased following a pattern similar to that of the adjuvanted Zika MP vaccine ODF. Adjuvanted Zika MP vaccine ODF produced significant IgG3 titers during weeks 2–11. The data are presented as the means ± SEMs; Brown–Forsythe ANOVA test followed by Tukey’s multiple comparison test; ns, nonsignificant, *, *p* < 0.05, **, *p* < 0.01, ***, *p* < 0.001, ****, *p* < 0.0001.

**Figure 13 vaccines-12-00720-f013:**
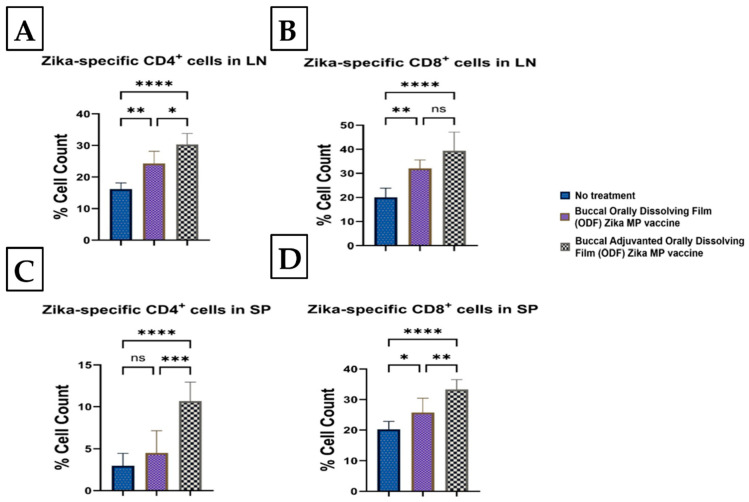
Helper CD4+ helper and CD8+ cytotoxic T cells were analyzed following a prime dose at week 0 and two booster doses at weeks 2 and 4. Zika-specific cellular T-cell responses following buccal immunization using ODFs were measured via flow cytometry [2]. In Figure 13, T-cell responses are shown for lymph nodes (CD4+ helper (**A**) and CD8+ cytotoxic T cells (**B**)) and spleen (CD4+ helper (**C**) and CD8+ cytotoxic (**D**)). The vaccine induced significant CD4+ helper and CD8+ cytotoxic T-cell surface marker cellular responses after buccal vaccination. Lymph nodes elicited a significant percentage of CD4+ helper and CD8+ cytotoxic T cells. In the spleen, the vaccine, with or without adjuvants, produced significantly higher CD8+ cytotoxic T cells than CD4+ T cells. The data are presented as the means ± SEMs; Brown–Forsythe ANOVA test followed by Tukey’s multiple comparison test; ns, nonsignificant; *, *p* < 0.05; **, *p* < 0.01; ***, *p* < 0.001; ****, *p* < 0.0001.

**Figure 14 vaccines-12-00720-f014:**
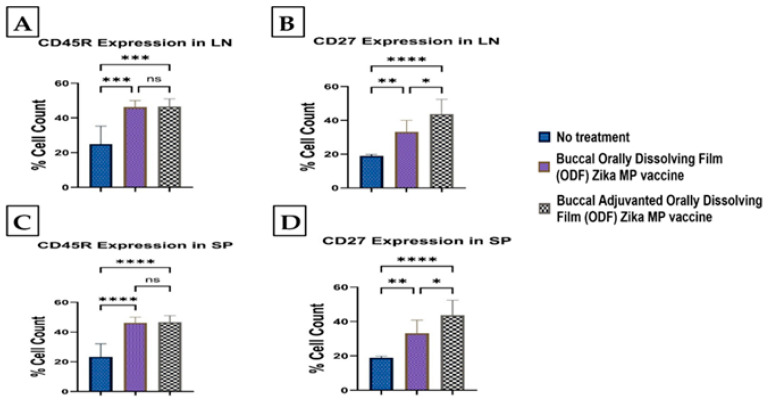
Flow cytometry was utilized to measure the Zika-specific expression of CD45R (**A**) and CD27 (**B**) memory markers in lymph nodes, as well as the Zika-specific expression of CD45R (**C**) and CD27 (**D**) T cells in the spleen. Swiss Webster mice received a prime dose at week 0, followed by two booster doses (at week 2 and week 4). The buccal ODF Zika MP vaccine and the adjuvanted ODF Zika MP vaccine elicited a significant CD45R and CD27 memory response as shown by the percent cell count following buccal vaccination. Both CD27 and CD45R markers were present in the lymph nodes, while both memory markers were detected in the spleen as well. (Data presented as Mean ± SEM, analyzed using Brown–Forsythe ANOVA test followed by Tukey’s multiple comparison test; ns, non-significant, *, *p* < 0.05, **, *p* < 0.01, ***, *p* < 0.001, ****, *p* < 0.0001).

**Figure 15 vaccines-12-00720-f015:**
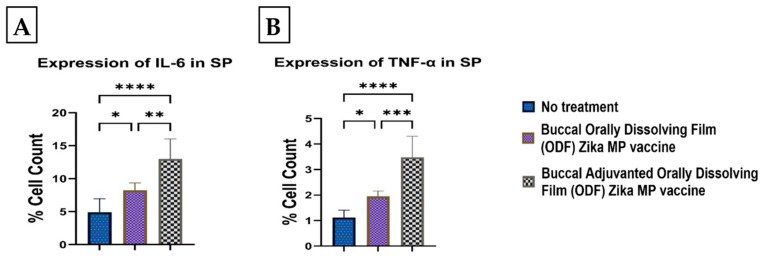
Zika-specific expression of intracellular markers Interleukin-6 (IL-6) (**A**) and Tumor Necrosis Factor alpha (TNF-α) (**B**) in the spleen. There was a significantly higher expression of intracellular cytokines IL-6 and TNF-α than in the no-treatment group in the spleen. Expression of IL-6 was significantly higher (fourfold higher) than the TNF-α cytokine. Adjuvanted Zika MP vaccine ODF induced significantly higher IL-6 and TNF-α cytokines than the unadjuvanted and naïve or no treatment group. Data represented as Mean ± SEM, Brown–Forsythe ANOVA test followed by Dunnett’s multiple comparison test; *, *p* < 0.1, **, *p* < 0.01, ***, *p* < 0.001, ****, *p* < 0.0001.

## Data Availability

The data presented in this study is available upon request from the corresponding author.

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
