# Peer review of "Buccal Administration of a Zika Virus Vaccine Utilizing 3D-Printed Oral Dissolving Films in a Mouse Model"

_vaccines, 2024, doi:10.3390/vaccines12070720_

Round 1

Reviewer 1 Report

Comments and Suggestions for Authors

The authors describe their in-vitro and in-vivo investigation of a candidate adjuvated and un-adjuvated inactivated microparticulate (MP) Zika vaccine intended for the prevention of complications such as microencephaly and Guillain–Barré syndrome (GBS). The goal was to characterize and evaluate a microparticle formulation that encapsulates the whole lyophilized inactivated Zika virus in polymeric MP that preserved the viral antigenic epitopes. The MP vaccine, encapsulating inactivated Zika virus, was embedded in a 3D-printed Orally Dissolving Film (ODF) for buccal delivery. After extensive formulation and characterization the immunostimulatory potential of vaccine MPs loaded in ODFs was evaluated in-vitro. Subsequently, experiments were performed to evaluate the immune response of the vaccine by administering  to mice via the buccal route with the help of thin films or oral dissolving films (ODFs). Mice were giving a prime dose and two booster doses with and without an adjuvant two weeks apart. The results of the in vitro experiments indicated that the ODFs had excellent physiochemical properties, thus indicating that the films were good carriers for vaccine microparticles and were biocompatible with the buccal mucosa. The results of the in vivo experiments demonstrated a humoral (IgG, subtypes IgG1 and IgG2a) and T-cell response (CD4+/CD8+) for ZIKV immunity. Both the Zika MP vaccine and the adjuvanted Zika MP vaccine elicited a memory (CD45R/CD27) response and intracellular cytokine (TNF-α and IL-6) expression. The authors concluded that the Zika MP candidate vaccine administered by the  buccal route with the aid of ODFs was promising for the development of pain-free vaccines for infectious diseases. As a proof of concept study, the rationale, approach and methods were appropriate and scientifically sound for demonstrating the feasibility of formulating an intranasal Zika microparticulate vaccine capable of eliciting both a humoral and cellular immune response. Overall, this is an excellently written and easily understood manuscript and the content are highly appropriate for the journal. The only weakness is in the experimental design that is lacking as indicated by the authors, the need to test the serum samples for virus neutralizing activity as a measurement of protective efficacy. Furthermore, a better estimate of ELISA antibody titers would have been actual numeric titers and not based on optimal density values. While the challenge experiment would have shed light on the potential neutralizing capacity, and thus the protectiveness of the antibody in the serum sample, the fact that a lethal dose of virus was not employed in the challenge experiment failed to address this critical aspect of vaccine efficacy.  The other minor point is that reference is made to antibody detected by an ELISA as being virus specific which is not correct as antibody detected by an ELISA are broadly cross reactive (Line 255). Finally, why were multiple doses, including a prime dose, and 2 subsequent booster doses evaluated rather than a single dose of the vaccine? While the likely answer is because of an inactivated formulated vaccine, the requirement of 3 doses of a vaccine is not likely to be challenging logistically.

Specific comments.

Lines 17 -18. In the abstract, the following sentence should use  were instead of are as past tense,  “In vitro, the ODFs displayed excellent physiochemical properties, indicating that the films are good carriers for vaccine microparticles and are biocompatible with the buccal mucosa”.

Figure 3 shows the timeline in days and in the results, the timeline is described in weeks, could be improved by presenting timeline data in only one timeframe

Comments on the Quality of English Language

As indicated by one of the reviewer's specific comments, contents need to be reviewed to ensure correct use of past tense

Author Response

Thank you for reviewing our manuscript. Please see the attached. 

Review 1 responses:

The authors describe their in-vitro and in-vivo investigation of a candidate adjuvated and un-adjuvated inactivated microparticulate (MP) Zika vaccine intended for the prevention of complications such as microencephaly and Guillain–Barré syndrome (GBS). The goal was to characterize and evaluate a microparticle formulation that encapsulates the whole lyophilized inactivated Zika virus in polymeric MP that preserved the viral antigenic epitopes. The MP vaccine, encapsulating inactivated Zika virus, was embedded in a 3D-printed Orally Dissolving Film (ODF) for buccal delivery. After extensive formulation and characterization the immunostimulatory potential of vaccine MPs loaded in ODFs was evaluated in-vitro. Subsequently, experiments were performed to evaluate the immune response of the vaccine by administering  to mice via the buccal route with the help of thin films or oral dissolving films (ODFs). Mice were giving a prime dose and two booster doses with and without an adjuvant two weeks apart. The results of the in vitro experiments indicated that the ODFs had excellent physiochemical properties, thus indicating that the films were good carriers for vaccine microparticles and were biocompatible with the buccal mucosa. The results of the in vivo experiments demonstrated a humoral (IgG, subtypes IgG1 and IgG2a) and T-cell response (CD4+/CD8+) for ZIKV immunity. Both the Zika MP vaccine and the adjuvanted Zika MP vaccine elicited a memory (CD45R/CD27) response and intracellular cytokine (TNF-α and IL-6) expression. The authors concluded that the Zika MP candidate vaccine administered by the  buccal route with the aid of ODFs was promising for the development of pain-free vaccines for infectious diseases. As a proof of concept study, the rationale, approach and methods were appropriate and scientifically sound for demonstrating the feasibility of formulating an intranasal Zika microparticulate vaccine capable of eliciting both a humoral and cellular immune response. Overall, this is an excellently written and easily understood manuscript and the content are highly appropriate for the journal.

Response: Thank you for reviewing the manuscript.

The only weakness is in the experimental design that is lacking as indicated by the authors, the need to test the serum samples for virus neutralizing activity as a measurement of protective efficacy. Furthermore, a better estimate of ELISA antibody titers would have been actual numeric titers and not based on optimal density values.

Response: Thank you for the suggestion. In this paper, we included the OD values as this is acceptable for publication. Below are a few papers from our lab and other labs that have recently been published using OD values. However, we will definitely take that into consideration for the next Zika manuscript.

Past publications from our lab:

  1. Vijayanand S, Patil S, Joshi D, Menon I, Braz Gomes K, Kale A, Bagwe P, Yacoub S, Uddin MN, D'Souza MJ. Microneedle Delivery of an Adjuvanted Microparticulate Vaccine Induces High Antibody Levels in Mice Vaccinated against Coronavirus. Vaccines (Basel). 2022 Sep 7;10(9):1491. doi: 10.3390/vaccines10091491. PMID: 36146568; PMCID: PMC9503342.
  2. Gomes KB, Menon I, Bagwe P, Bajaj L, Kang SM, D'Souza MJ. Enhanced Immunogenicity of an Influenza Ectodomain Matrix-2 Protein Virus-like Particle (M2e VLP) Using Polymeric Microparticles for Vaccine Delivery. Viruses. 2022 Aug 30;14(9):1920. doi: 10.3390/v14091920. PMID: 36146733; PMCID: PMC9506217.

Publications from other labs:

  1. Maltseva M, Galipeau Y, Renner TM, Deschatelets L, Durocher Y, Akache B, Langlois MA. Characterization of Systemic and Mucosal Humoral Immune Responses to an Adjuvanted Intranasal SARS-CoV-2 Protein Subunit Vaccine Candidate in Mice. Vaccines (Basel). 2022 Dec 23;11(1):30. doi: 10.3390/vaccines11010030. PMID: 36679875; PMCID: PMC9865305.
  2. Muranishi K, Kinoshita M, Inoue K, Ohara J, Mihara T, Sudo K, Ishii KJ, Sawa T, Ishikura H. Antibody Response Following the Intranasal Administration of SARS-CoV-2 Spike Protein-CpG Oligonucleotide Vaccine. Vaccines (Basel). 2023 Dec 20;12(1):5. doi: 10.3390/vaccines12010005. PMID: 38276664; PMCID: PMC10818492.

While the challenge experiment would have shed light on the potential neutralizing capacity, and thus the protectiveness of the antibody in the serum sample, the fact that a lethal dose of virus was not employed in the challenge experiment failed to address this critical aspect of vaccine efficacy.  

Response: Thank you for the suggestion. We definitely agree that if the lethal dose was given, then this would have unraveled the potential of neutralizing capacity and the protection of the antibody in serum. In our next follow up study, we will determine LD50 then will administer a lethal dose of virus.

The other minor point is that reference is made to antibody detected by an ELISA as being virus specific which is not correct as antibody detected by an ELISA are broadly cross reactive (Line 255).

Response: Thank you for the suggestion. We have changed line 255.

Finally, why were multiple doses, including a prime dose, and 2 subsequent booster doses evaluated rather than a single dose of the vaccine? While the likely answer is because of an inactivated formulated vaccine, the requirement of 3 doses of a vaccine is not likely to be challenging logistically.

Response: Thank you for the suggestion. Since we used the inactivated formulation approach, an additional booster was required. In other reports, a phase I study published, they found that a third dose of Zika vaccine overcame discrepancies in immunogenicity, and thus warranting a three-dose vaccine approach (https://doi.org/10.1016/S1473-3099(23)00192-5). However, in order to reduce the number of dose, we will plan to investigate a two dose approach along with adjuvants in our follow up paper.

Specific comments.

Lines 17 -18. In the abstract, the following sentence should use  were instead of are as past tense,  “In vitro, the ODFs displayed excellent physiochemical properties, indicating that the films are good carriers for vaccine microparticles and are biocompatible with the buccal mucosa”.

Response: Thank you. We have changed this sentence.

Figure 3 shows the timeline in days and in the results, the timeline is described in weeks, could be improved by presenting timeline data in only one timeframe

Response: Thank you for the suggestion. We have changed figure 3 to represent weeks, so it is consistent.

Reviewer 2 Report

Comments and Suggestions for Authors

In their article untitled « buccal administration of a Zika virus vaccine utilizing 3D-Printed oral dissolving films (ODFs) in a mouse model » Shah et al. described the preparation and first in vitro (on dendritic cells) and in vivo assays (in swiss-webster mice) of an oral dissolving films containing nanoparticles encapsided inactivated Zika virus (iZIKV) with or without adjuvant (alum and MPL-A).

The work is interesting as it provided a new interesting way of vaccination with a quite robust system. However, as it appears in the description of the ODFs preparation, I’m not sure that this process can go up to a large-scale production ie thousand or thousands of thousand individual ODF. But this is not the purpose of this study, I agree.

The description of the physical and biochemical properties of the ODFs is also clean and convincing except there are no demonstration of stability of the prepared ODFs even if it is claimed line 617 in the discussion that stability is a crucial issue. Even if no cold-chain is required, in countries were ZIKA epidemic occurs, weather and local condition may exposed the material up to 30-35°C.

The in vivo assay shows a robust immune response (in term of antibodies) in the mice receiving the ODFs with iZIKV and adjuvant. The immunisation data are clear but the timing is confusing see below the detail requested.

Morevover the choice of the Swiss-Webster mice is not really accurate for a challenge assay as it is recognized that these mice are not sensitive to the infection by ZIKV, there are only a small replication not detectable in the large majority of the animals as noted in many articles (see review from Balint et al. in Viruses. 2021 Nov 8;13(11):2244. doi: 10.3390/v13112244.).

This limitation should be discussed and the choice of this model more clearly explained.

Major comments:

Within the introduction, why did you not speech about the large epidemic in Brazil then in South America line 38 and 39 ? It was during these epidemic that researcher community and public were aware of the congenital defect that remained anecdotal in the 2013 and 2014 epidemic in Pacific Ocean islands.  

The timing of the in vivo assay is confusing, you provide a time scale in day but described the sampling time in days and there are no correlation between those.

In the figure 3 there is D0 (Week 0?),  day 15 (week 2,), day 29 (week 4 day 1), day 77 (week 11) for the challenge and day 84 (week 12) for the immune organs collection.

Please correct and take in account the following points.

During the description you speech in figure about week 8 and 11 but there is no information if the animals were bleed before or after the challenge ! We all recognize that in this model the viral replication is limited and even cannot be assessed directly in many mice. But yes if there is an anamnestic response the week after this may sign the detection of the replicating virus by the immune system of the mice.

To note in figure 9 at week 0 (before prime ?) there are a difference in IgG detected between the 3 groups ! Please comment as it is not done, is there non-specific signal ? it should be indicated a cut-off for specific response within the following week. There are also some detection of IgG in non-treated group at week 4, 6 8 and 11. Is this detection in the non-treated group due to manipulation of the mice ?

In addition, line 440 you said that IgA increased post challenge at week 11. I’m pretty sure the it cannot be any anamnestic effect of the challenge if you bleed the animal the same day (or just before as it as to be done).

Minor comments:

The table (not a figure) is duplicated line 317-138 page 8

Line 401: If you use significant please provide a p values (ie do a count of the autophagosomes in the different conditions).

Author Response

Thank you for reviewing our manuscript. Please see attached. 

Review 2 response:

In their article untitled « buccal administration of a Zika virus vaccine utilizing 3D-Printed oral dissolving films (ODFs) in a mouse model » Shah et al. described the preparation and first in vitro (on dendritic cells) and in vivo assays (in swiss-webster mice) of an oral dissolving films containing nanoparticles encapsided inactivated Zika virus (iZIKV) with or without adjuvant (alum and MPL-A). The work is interesting as it provided a new interesting way of vaccination with a quite robust system. However, as it appears in the description of the ODFs preparation, I’m not sure that this process can go up to a large-scale production ie thousand or thousands of thousand individual ODF. But this is not the purpose of this study, I agree.

The description of the physical and biochemical properties of the ODFs is also clean and convincing except there are no demonstration of stability of the prepared ODFs even if it is claimed line 617 in the discussion that stability is a crucial issue. Even if no cold-chain is required, in countries were ZIKA epidemic occurs, weather and local condition may exposed the material up to 30-35°C.

 Response: Thank you for the comment. We agree that stability is an important factor since the weather and local condition may be different depending on the location. In our films, stability testing was performed as per International Council on Harmonization (ICH) guidelines under several conditions. Three film strips from each formulation were wrapped with butter paper and then with aluminum foil to be stored in the stability chamber. These films were stored under two storage conditions: 25°C/60% relative humidity (RH) (Zone II) and at 40°C/75% R.H. (zone IVb). Stored film strips were sampled at varying intervals (0, 1 week, 1 month, 3 months, 6 months and 12 months) for further testing for overall appearance, physiochemical properties, and weight variation. We found that even at 12 months, films retained their physiochemical properties, and films’ morphology did not change much. In addition, in the next our next ODF manuscript, we will plan to do more detailed stability studies of heat capacities such as DSC since this this can provide more information on the stability of the ODF.

The in vivo assay shows a robust immune response (in term of antibodies) in the mice receiving the ODFs with iZIKV and adjuvant. The immunization data are clear, but the timing is confusing see below the detail requested.

 Response: Thank you for the comment. We have changed Figure 3 timeline to represent weeks, so it is consistent now.

Morevover the choice of the Swiss-Webster mice is not really accurate for a challenge assay as it is recognized that these mice are not sensitive to the infection by ZIKV, there are only a small replication not detectable in the large majority of the animals as noted in many articles (see review from Balint et al. in Viruses. 2021 Nov 8;13(11):2244. doi: 10.3390/v13112244.). This limitation should be discussed and the choice of this model more clearly explained.

 Response: Thank you for the comment. In this proof-of-concept study, we used swiss-webster mice for testing a buccal vaccine for Zika. However, one of the limitations as mentioned in the paper was the robustness of the viral challenge. Other labs groups have used the swiss-webster mice for Zika studies. However, we plan on testing our vaccine using other mice that may be more sensitive to the infection by ZIKV in our next Zika manuscript.

Publications from other labs:

  1. Adam, A., Fontes-Garfias, C.R., Sarathy, V.V. et al.A genetically stable Zika virus vaccine candidate protects mice against virus infection and vertical transmission. npj Vaccines 6, 27 (2021). https://doi.org/10.1038/s41541-021-00288-6
  2. Charles B. Stauft, Oleksandr Gorbatsevych, Jeronimo Cello, Eckard Wimmer, Bruce Futcher. Comparison of African, Asian, and American Zika Viruses in Swiss Webster mice: Virulence, neutralizing antibodies, and serotypes. bioRxiv 075747, 2016. doi: https://doi.org/10.1101/075747.

Major comments:

Within the introduction, why did you not speech about the large epidemic in Brazil then in South America line 38 and 39 ? It was during these epidemic that researcher community and public were aware of the congenital defect that remained anecdotal in the 2013 and 2014 epidemic in Pacific Ocean islands.  

 Response: Thank you for the comment. We have added the following in the paper: “The significant outbreak in Brazil, with over 100,000 reported cases of Zika, as well as its spread in South America, drew attention from the research community and raised public awareness regarding the congenital defect that had previously been considered anecdotal during the 2013 and 2014 epidemic in Pacific Ocean islands.”

The timing of the in vivo assay is confusing, you provide a time scale in day but described the sampling time in days and there are no correlation between those. In the figure 3 there is D0 (Week 0?),  day 15 (week 2,), day 29 (week 4 day 1), day 77 (week 11) for the challenge and day 84 (week 12) for the immune organs collection.

Please correct and take in account the following points.

 Response: Thank you for the comment. We have changed figure 3 timeline to represent weeks, so it is consistent.

During the description you speech in figure about week 8 and 11 but there is no information if the animals were bleed before or after the challenge ! We all recognize that in this model the viral replication is limited and even cannot be assessed directly in many mice. But yes if there is an anamnestic response the week after this may sign the detection of the replicating virus by the immune system of the mice.

Response: Thank you for the comment. The mice were bleed at week 8 (before challenge), then at week 10 the challenge was done. After 1 week of exposure period for virus to replicate in mice, the mice were bleed after challenge, during sacrifice at week 11.

To note in figure 9 at week 0 (before prime ?) there are a difference in IgG detected between the 3 groups ! Please comment as it is not done, is there non-specific signal ? it should be indicated a cut-off for specific response within the following week. There are also some detection of IgG in non-treated group at week 4, 6 8 and 11. Is this detection in the non-treated group due to manipulation of the mice ?

Response: Thank you for the comment. Before the prime dose was give, serum was collected. Since the mice were kept in the same room and maybe to handing of the mice to collect serum, mice may have undergone stress triggers. This is consistent with other labs where they observed some response in antibodies.

  1. https://doi.org/10.1007/s40744-022-00433-0
  2. doi: 1016/j.celrep.2021.109595

In addition, line 440 you said that IgA increased post challenge at week 11. I’m pretty sure the it cannot be any anamnestic effect of the challenge if you bleed the animal the same day (or just before as it as to be done).

Response: Thank you for the comment. In week 10, mice were challenged. After one week, at week 11, the mice were sacrificed. Post challenge refers to after the one week of exposure of the Zika virus.

Minor comments:

The table (not a figure) is duplicated line 317-138 page 8

Response: Thank you. We have corrected this in figure 4.

Line 401: If you use significant please provide a p values (ie do a count of the autophagosomes in the different conditions).

Response: Thank you for the comment. We have added p values in the caption of figure 8.
